# AN INFORMATION-THEORETIC METRIC OF TRANSFERABILITY FOR TASK TRANSFER LEARNING

## ABSTRACT

An important question in task transfer learning is to determine task transferability, i.e. given a common input domain, estimating to what extent representations learned from a source task can help in learning a target task. Typically, transferability is either measured experimentally or inferred through task relatedness, which is often defined without a clear operational meaning. In this paper, we present a novel metric, *H-score*, an easily-computable evaluation function that estimates the performance of transferred representations from one task to another in classification problems. Inspired by a principled information theoretic approach, H-score has a direct connection to the asymptotic error probability of the decision function based on the transferred feature. This formulation of transferability can further be used to select a suitable set of source tasks in task transfer learning problems or to devise efficient transfer learning policies. Experiments using both synthetic and real image data show that not only our formulation of transferability is meaningful in practice, but also it can generalize to inference problems beyond classification, such as recognition tasks for 3D indoor-scene understanding.

## 1 INTRODUCTION

*Transfer learning* is a learning paradigm that exploits relatedness between different learning tasks in order to gain certain benefits, e.g. reducing the demand for supervision (Pratt (1993)). In *task transfer learning*, we assume that the input domain of the different tasks are the same. Then for a target task $\mathcal{T}_T$, instead of learning a model from scratch, we can initialize the parameters from a previously trained model for some related source task $\mathcal{T}_S$. For example, deep convolutional neural networks trained for the ImageNet classification task have been used as the source network in transfer learning for target tasks with fewer labeled data (Donahue et al. (2014)), such as medical image analysis (Shie et al. (2015)) and structural damage recognition in buildings (Gao & Mosalam).

An imperative question in task transfer learning is *transferability*, i.e. *when a transfer may work* and *to what extent*. Given a metric capable of efficiently and accurately measuring transferability across arbitrary tasks, the problem of task transfer learning, to a large extent, is simplified to search procedures over potential transfer sources and targets as quantified by the metric. Traditionally, transferability is measured purely empirically using model loss or accuracy on the validation set (Yosinski et al. (2014); Zamir et al. (2018); Conneau et al. (2017)). There have been theoretical studies that focus on *task relatedness* (Baxter (2000); Maurer (2009); Pentina & Lampert (2014); Ben-David et al. (2003)). However, they either cannot be computed explicitly from data or do not directly explain task transfer performance. In this study, *we aim to estimate transferability analytically, directly from the training data*.

We quantify the transferability of feature representations across tasks via an approach grounded in statistics and information theory. The key idea of our method is to show that the error probability of using a feature of the input data to solve a learning task can be characterized by a linear projection of this feature between the input and output domains. Hence we adopt the projection length as a metric of the feature's effectiveness for the given task, and refer to it as the *H-score* of the feature. More generally, H-score can be applied to evaluate the performance of features in different tasks, and is particularly useful to quantify feature transferability among tasks. Using this idea, we define *task transferability* as the normalized H-score of the optimal source feature with respect to the target task.

As we demonstrate in this paper, the advantage of our transferability metric is threefold. (i) it has a strong operational meaning rooted in statistics and information theory; (ii) it can be computed directly and efficiently from the input data, with fewer samples than those needed for empirical learning; (iii) it can be shown to be strongly consistent with empirical transferability measurements.

In this paper, we will first present the theoretical results of the proposed transferability metric in Section 2-4. Section 5 presents several experiments on real image data , including image classificaton tasks using the Cifar 100 dataset and 3D indoor scene understanding tasks using the Taskonomy dataset created by Zamir et al. (2018). A brief review of the related works is included in Section 6.

## 2   BACKGROUND

In this section, we will introduce the notations used throughout this paper, as well as some related concepts in Euclidean information theory and statistics.

$X, x, \mathcal{X}$ and $P_X$ represent a random variable, a value, the alphabet and the probability distribution respectively. $\sqrt{P_X}$ denotes the vector with entries $\sqrt{P_X(x)}$ and $[\sqrt{P_X}]$ the diagonal matrix of $\sqrt{P_X(X)}$. For joint distribution $P_{YX}$, $P_{YX}$ represents the $|\mathcal{Y}| \times |\mathcal{X}|$ probability matrix. Depending on the context, $f(X)$ is either a $|\mathcal{X}|$-dimensional vector whose entries are $f(x)$, or a $|\mathcal{X}| \times k$ feature matrix. Further, we define a task to be a tuple $\mathcal{T} = (X, Y, P_{XY})$, where $X$ is the training features and $Y$ is the training label, and $P_{XY}$ the joint probability (possibly unknown). Subscripts $S$ and $T$ are used to distinguish the source task from the target task.

### 2.1   LOCAL INFORMATION GEOMETRY AND HYPOTHESIS TESTING

Our definiton of transferability uses concepts in local information geometry developed by Makur et al. (2015), which characterizes probability distributions as vectors in the information space. Consider the following binary hypothesis testing problem: test whether i.i.d. samples $\mathbf{x}^m = \{x^{(i)}\}_{i=1}^m$ are drawn from distribution $P_1$ or distribution $P_2$, where $P_1, P_2$ belong to an $\epsilon$-neighborhood $\mathcal{N}_\epsilon(P_0) \triangleq \{P | \sum_{x \in \mathcal{X}} \frac{(P(x) - P_0(x))^2}{P_0(x)} \leq \epsilon^2\}$ centered at a reference distributon $P_0$. And denote $\phi_i = \frac{P_i(x) - P_0(x)}{\epsilon \sqrt{P_0(x)}}$ as the *information vector* corresponding to $P_i$ for $i = 1, 2$.

Given $k$ normalized feature functions $f(x) = [f_1(x), \ldots, f_k(x)]$, let $\Xi(x) = [\xi_1(x), \ldots, \xi_k(x)]$ be the matrix of *feature vectors* $\xi_i(x) = \sqrt{P_0(x)} f_i(x)$. Given observations $\mathbf{x}^m$, feature $f$ is associated with a k-dimensional statistics $l(\mathbf{x}^m) = \frac{1}{m} \sum_{i=1}^m (f(x^{(i)}))$ for the binary hypothesis testing problem. Let $E_f$ be the error exponent of decision region $\{\mathbf{x}^m \mid l(\mathbf{x}^m) > T\}$ for $T \geq 0$, which characterizes the asymptotic error probability $P_e$ of $l$ (i.e. $\lim_{m \to \infty} -\frac{1}{m} \log(P_e) = E_f^k$). $E_f$ can be written as the squared length of a projection:

$$E_f^k = \sum_{i=1}^k E_{f_i} = \sum_{i=l}^k \frac{\epsilon^2}{8} \langle \xi_i, \phi_1 - \phi_2 \rangle^2 + o(\epsilon^2) \tag{1}$$

When $f(x) = \log \frac{P_1(x)}{P_2(x)}$ is the log likelihood ratio, $l$ is the minimum sufficient statistics that achieves the largest error exponent $E_f^k = \sum_{i=1}^k \frac{\epsilon^2}{8} ||\phi_1 - \phi_2||^2 + o(\epsilon^2)$ by the Chernoff theorem. (See Appendix A for details.) In the rest of this paper, we assume $\epsilon$ is small.

### 2.2   DIVERGENCE TRANSITION MATRIX

**Definition 1.** *Matrix $\tilde{B}$ is the* Divergence Transition Matrix (DTM) *of a joint probability $P_{YX}$ if* $\tilde{B} = [\sqrt{P_Y}]^{-1} P_{YX} [\sqrt{P_X}]^{-1} - \sqrt{P_Y} \sqrt{P_X}^\mathrm{T}$.

The singular values of $\tilde{B}$ satisfy that $1 \geq \sigma_1 \geq \cdots \geq \sigma_K = 0, K \triangleq \min\{|\mathcal{X}|, |\mathcal{Y}|\}$. Let $\Psi = [\psi_1, \ldots, \psi_K]$ and $\Phi = [\phi_1, \ldots, \phi_K]$ be the left and right singular vectors of $\tilde{B}$. Define functions $f_i^*(x) = \frac{\phi_i(x)}{\sqrt{P_X(x)}}$ and $g_i^*(y) = \frac{\psi_i(y)}{\sqrt{P_Y(y)}}$ for each $i = 1, \ldots, K - 1$. Makur et al. (2015) further

proved that $f_i^*$ and $g_i^*$ are solutions to the *maximal HGR correlation problem* studied by Hirschfeld (1935); Gebelein (1941); Rényi (1959), defined as follows:

$$\rho(X;Y) = \sup_{\substack{f: \mathcal{X} \to \mathbb{R}^k, g: \mathcal{Y} \to \mathbb{R}^k \\ \mathbb{E}[f(X)] = \mathbb{E}[g(Y)] = 0 \\ \mathbb{E}[f(X)f(X)^T] = I}} \mathbb{E}[f(X)^T g(Y)] \tag{2}$$

The maximal HGR problem finds the $K$ strongest, independent modes in $P_{XY}$ from data. It can be solved efficiently using the Alternating Conditional Expectation (ACE) algorithm with provable error bound (see Appendix B). Huang et al. (2017) further showed that $f^*$ and $g^*$ are the *universal minimum error probability features* in the sense that they can achieve the smallest error probability over all possible inference tasks.

## 3 MEASURING THE EFFECTIVENESS OF FEATURES IN CLASSIFICATION TASKS

In this section, we present a performance metric of a given feature representation for a learning task.

### 3.1 H-SCORE

For a classification task involving input variable $X$ and label $Y$, most learning algorithms work by finding a $k$-dimensional functional representation $f(x)$ of $X$ that is most discriminative for the classification. To measure how effective $f(x)$ is in predicting $Y$, rather than train the model via gradient descent and evaluate its accuracy, we present an analytical approach based on the definition below:

**Definition 2.** *Given data matrix $X \in \mathbb{R}^{m \times d}$ and label $Y \in \{1, \ldots, |\mathcal{Y}|\}$. Let $f(x)$ be a $k$-dimensional, zero-mean feature function. The H-Score of $f(x)$ with respect to the learning task represented by $P_{YX}$ is:*

$$\mathcal{H}(f) = \mathrm{tr}(\mathrm{cov}(f(X))^{-1}\mathrm{cov}(\mathbb{E}_{P_{X|Y}}[f(X)|Y]))$$

This definition is intuitive from a nearest neighbor search perspective. i.e. a high H-score implies the inter-class variance $\mathrm{cov}(\mathbb{E}_{P_{X|Y}}[f(X)|Y])$ of $f$ is large, while feature redundancy $\mathrm{tr}(\mathrm{cov}(f(X)))$ is small. Such a feature is discriminative and efficient for learning label Y. More importantly, $\mathcal{H}(f)$ has a deeper operational meaning related to the asymptotic error probability for a decision rule based on $f$ in the hypothesis testing context, discussed in the next section.

### 3.2 OPERATIONAL MEANING OF H-SCORE

Without loss of generality, we consider the binary classification task as a hypothesis testing problem defined in Section 2.1, with $P_1 = P_{X|Y=0}$, and $P_2 = P_{X|Y=1}$. For any k-dimensional feature representation $f(x)$, we can quantify its performance with respect to the learning task using its error exponent $E_f^k$.

**Theorem 1.** *Given $P_{X|Y=0}, P_{X|Y=1} \in \mathcal{N}_\epsilon^{\mathcal{X}}(P_{0,X})$ and features $f$ such that $\mathbb{E}[f(X)] = 0$ and $\mathbb{E}[f(X)f(X)^T] = I$, there exists some constant $c$ independent of $f$ such that $E_f^k = c\mathcal{H}(f)$.*

See Appendix C for the proof. The above theorem shows that H-score $\mathcal{H}(f)$ is proportional to the error exponent of the decision region based on $f(x)$ when $f(x)$ is zero-mean with identity covariance. To compute the H-score of arbitrary $f$, we can center the features $f_S(x) - \mathbb{E}[f_S(x)]$, and incorporate normalization into the computation of the error exponent, which results in Definition 2. The details are presented in Appendix D.

The proof for Theorem 1 uses the fact that $\mathcal{H}(f) = \|\tilde{B}\Xi\|_F^2$, where $\tilde{B}$ is the DTM matrix, $\Xi \triangleq [\xi_1 \cdots \xi_k]$ is the matrix composed of information vectors $\xi_i$ and $c$ is a constant. This allows us to infer an upper bound for the H-score of a given learning task:

**Corollary 1.** *For all normalized feature $f(x) = [f_1(x), \ldots, f_k(x)]$, its H-score satisfies $\mathcal{H}(f) \leq \sum_{i=1}^k \sigma_i^2 \leq k$, where $\sigma_i$ is the ith singular value of $\tilde{B}$.*

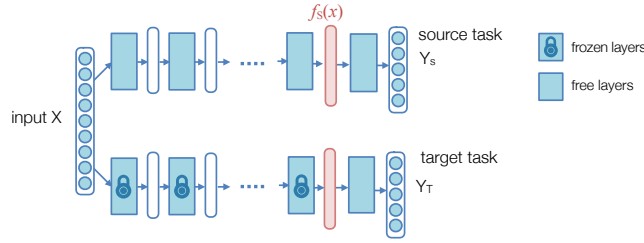

Figure 1: A simple neural network topology for transfer learning

The first inequality is achieved when $\Xi$ is composed of the right singular vectors of $\tilde{B}$, i.e. $\max_\Xi ||\tilde{B}\Xi||_F^2 = ||\tilde{B}_T||_F^2$. The corresponding feature functions $\mathrm{f}_{\mathrm{opt}}(X)$is in fact the same as the universal minimum error probability features from the maximum HGR correlation problem. The final inequality in Corollary 1 is due to the fact all singular values of $\tilde{B}$ are less than or equal to 1.

## 4 TRANSFERABILITY

Next, we apply H-score to efficiently measure the effectiveness of task transfer learning. We will also discuss how this approach can be used to solve the source task selection problem.

### 4.1 TASK TRANSFERABILITY

A typical way to transfer knowledge from the source task $\mathcal{T}_S$ to target task $\mathcal{T}_T$ is to train the target task using source feature representation $f_S(x)$. In a neural network setting, this idea can be implemented by copying the parameters from the first N layers in the trained source model to the target model, assuming the model architecture on those layers are the same. The target classifier then can be trained while freezing parameters in the copied layers (Figure 1). Under this model, a natural way to quantify transferability is as follows:

**Definition 3** (Task transferability). *Given source task $\mathcal{T}_S$ and target task $\mathcal{T}_T$, and trained source feature representation $f_S(x)$, the* transferability *from $\mathcal{T}_S$ to $\mathcal{T}_T$ is $\mathfrak{T}(S,T) \triangleq \frac{\mathcal{H}_T(f_S)}{\mathcal{H}_T(f_{T_{\mathrm{opt}}})}$, where $f_{T_{\mathrm{opt}}}(x)$ is the minimum error probability feature of the target task.*

The statement $\mathfrak{T}(S,T) = r$ means the error exponent of transfering from $\mathcal{T}_S$ via feature representation $f_S$ is $\frac{1}{r}$ of the *optimal* error exponent for predicting the target label $Y_T$. This definition also implies $0 \leq \mathfrak{T}(S,T) \leq 1$, which satisfies the data processing inequality if we consider the transferred feature $f_S(X)$ as post-processing of input $X$ for solving the target task. And it can not increase information about predicting the target task $\mathcal{T}$.

A common technique in task transfer learning is fine-tuning, which adds before the target classifier additional free layers, whose parameters are optimized with respect to the target label. For the operational meaning of transferability to hold exactly, we require the fine tuning layers consist of only linear transformations, such as the model illustrated in Figure 2.a. It can be shown that under the local assumption, H-score is equivalent to the log-loss of the linear transfer model up to a constant offset (Appendix E). Nevertheless, later we will demonstrate empirically that this transferability metric can still be used for comparing the *relative* task transferability with fine-tuning.

### 4.2 EFFICIENT COMPUTATION OF TRANSFERABILITY

With a known $f$, computing H-score from $m$ sample data only takes $O(mk^2)$ time, where $k$ is the dimension of $f(x)$ for $k < m$. The majority of the computation time is spent on computing the sample covariance matrix $\mathrm{cov}(f(X))$.

The remaining question is how to obtain $f_{T_{opt}}$ efficiently. We use the fact that $\mathcal{H}_T(f_{\mathrm{opt}}) = ||\tilde{B}_T||_F^2 = \mathbb{E}[f(X)^{\mathrm{T}}g(Y)]$, where $f$ and $g$ are the solutions of the HGR-Maximum Correlation problem. This problem can be solved efficiently using the ACE algorithm for discrete variable $X$. For a continuous random variable $X$, we can obtain $f_{opt}$ through a different formulation of the HGR maximal

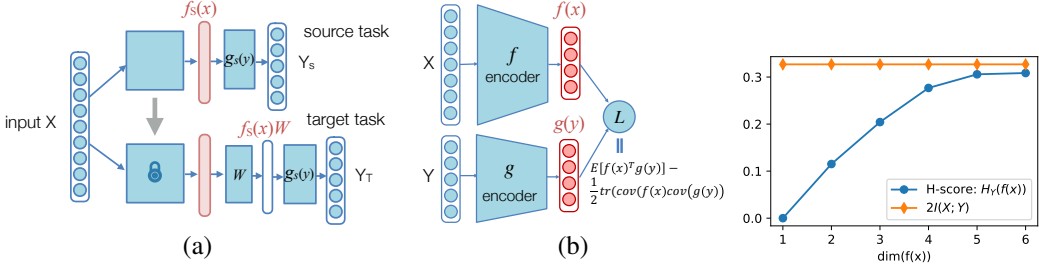

Figure 2: a) Network topology of linear feature transfer. b) Architecture for the neural network implementation of the soft HGR problem

Figure 3: H-score and mutual information.

correlation problem:

$$\max_{f:\mathcal{X}\to\mathbb{R}^k,g:\mathcal{Y}\to\mathbb{R}^k} \mathbb{E}[f(X)^{\mathrm{T}}g(Y)] - \frac{1}{2}\mathrm{tr}(\mathrm{cov}(f(X))\mathrm{cov}(g(Y))) \tag{3}$$
$$s.t. \quad \mathbb{E}[f(X)] = 0, \ \mathbb{E}[g(Y)] = 0$$

This is also known as the *soft HGR problem* studied by Wang et al. (2019), who reformulated the original maximal HGR correlation objective to eliminate the whitening constraints while having theoretically equivalent solution. In practice, we can utilize neural network layers to model functions $f$ and $g$, as shown in Figure 2.b. Given two branches of $k$ output units for both $f$ and $g$, the loss function can be evaluated in $O(mk^2)$, where $m$ is the batch size. Makur et al. (2015) showed that the sample complexity of ACE is only $1/k$ of the complexity of estimating $P_{Y,X}$ directly. This result also applies to the soft HGR problem due to their theoretical equivalence. Hence transferability can be computed with much less samples than actually training the transfer network.

It's also worth noting that, when $f$ is fixed, maximizing the objective in Equation 3 with respect to zero-mean function $g$ results in the definition of H-score. In many cases though, the computation of $H_T(f_{opt})$ can even be skipped entirely, such as the problem below:

**Definition 4** (Source task selection). *Given $N$ source tasks $\mathcal{T}_{S_1},\ldots,\mathcal{T}_{S_N}$ with labels $Y_{S_1},\ldots,Y_{S_N}$ and a target task $\mathcal{T}_T$ with label $Y_T$. Let $f_{S_1},\ldots,f_{S_N}$ be the minimum error probability feature functions of the source tasks. Find the source task $\mathcal{T}_{S_i}$ that maximizes the testing accuracy of predicting $Y_T$ with feature $f_{S_i}$.*

We can solve this problem by selecting the source task with the largest transferability to $\mathcal{T}_T$. In fact, we only need to compute the numerator in the transferability definition since the denominator is the same for all source tasks, i.e. $\mathrm{argmax}_i \mathfrak{T}(S_i, T) = \mathrm{argmax}_i(\mathcal{H}_T(f_{S_{i,\mathrm{opt}}})/\mathcal{H}_T(f_{T_{\mathrm{opt}}}) = \mathrm{argmax}_i \mathcal{H}(f_{S_i})$.

### 4.3 RELATIONSHIP WITH MUTUAL INFORMATION

Under the local assumption that $P_{X|Y} \in N_\epsilon(P_X)$, we can show that mutual information $I(X;Y) = \frac{1}{2}||B||_F^2 + o(\epsilon^2)$. (See Appendix F for details.) Hence H-score is related to mutual information by $\mathcal{H}(f(x)) \leq 2I(X;Y)$ for any zero-mean features $f(x)$ satisfying the aforementioned conditions.

Figure 3 compares the optimal H-score of a synthesized task when $|\mathcal{Y}| = 6$ with the mutual information between input and output variables, when the feature dimension $k$ changes. The value of H-score increases as $k$ increases, but reaches the upper bound when $k \geq 6$, since the rank of the joint probability between $X$ and $Y_T$, as well as the rank of its DTM is 6. As expected, the H-score values are below $2I(X;Y)$, with a gap due to the constant $o(\epsilon^2)$. This relationship shows that H-score is consistent with mutual information with a sufficiently large feature dimension $k$.

In practice, H-score is much easier to compute than mutual information when the input variable $X$ (or $f_S(X)$) is continous, as mutual information are either computed based on binning, which has extra bias due to bin size, or more sophisticated methods such as kernel density estimation or neural networks (Gabrié et al. (2018)). On the other hand, H-score only needs to estimate conditional expectations, which requires less samples.

## 5 EXPERIMENTS

In this section, we validate and analyze our transferability metric through experiments on real image data. [1] The tasks considered cover both object classification and non-classification tasks in computer vision, such as depth estimation and 3D (occlusion) edge detection.

### 5.1 VALIDATION OF TRANSFER PERFORMANCE

To demonstrate that our transferability metric is indeed a suitable measurement for task transfer performance, we compare it with the empirical performance of transfering features learned from ImageNet 1000-class classification (Krizhevsky et al. (2012)) to Cifar 100-class classification (Krizhevsky & Hinton (2009)), using a network similar to Figure 1. Comparing to ImageNet-1000, Cifar-100 has smaller sample size and its images have lower resolution. Therefore it is considered to be a more challenging task than ImageNet, making it a suitable case for transfer learning. In addition, we use a pretrained ResNet-50 as the source model due to its high performance and regular structure.

**Validation of H-score.** The training data for the target task in this experiemnt consists of $20,000$ images randomly sampled from Cifar-100. It is further split 9:1 into a training set and a testing set. The transfer network was trained using stochastic gradient descent with batch size $20,000$ for $100$ epochs.

Figure 4.a compares the H-score of transferring from five different layers (4a-4f) in the source network with the target log-loss and test accuracy of the respective features. As H-score increases, log-loss of the target network decreases almost linearly while the training and testing accuracy increase. Such behavior is consistent with our expectation that H-score reflects the learning performance of the target task. We also demonstrated that target sample size does not affect the relationship between H-score and log-loss (Figure 4.b).

This experiment also showed another potential application of H-score for selecting the most suitable layer for fine-tuning in transfer learning. In the example, transfer performance is better when an upper layer of the source networks is transferred. This could be because the target task and the source task are inherently similar such that the representation learned for one task can still be discriminative for the other.

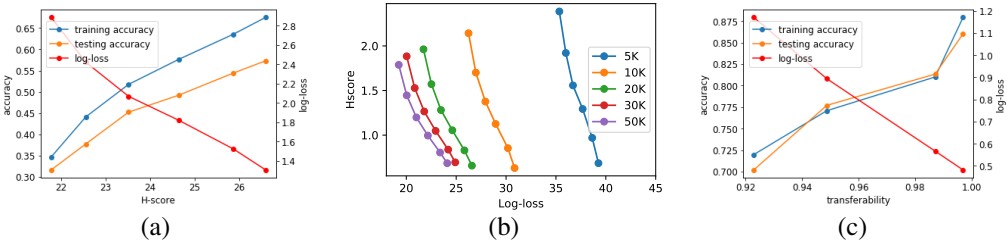

(a)  (b)  (c)

Figure 4: H-score and transferability vs. the empirical transfer performance measured by log-loss, training and testing accuracy. a.) Performance of ImageNet-1000 features from layers 4a-4f for Cifar-100 classification. b.) Effect of sample size (5K-50K) on H-score for Cifar 100. c.): Transferability from ImageNet-1000 to 4 different target tasks based on Cifar-100.

**Validation of Transferability.** We further tested our transferability metric for selecting the best target task for a given source task. In particular, we constructed 4 target classification tasks with 3, 5, 10, and 20 object categories from the Cifar-100 dataset. We then computed the transferability from ImageNet-1000 (using the feature representation of layer 4f) to the target tasks. The results are compared to the empirical transfer performance trained with batch size 64 for 50 epochs in Figure4.c. We observe a similar behavior as the H-score in the case of a single target task in Figure 4.a, showing that transferability can directly predict the empirical transfer performance.

### 5.2 TASK TRANSFER FOR 3D SCENE UNDERSTANDING

In this experiment, we solve the source task selection problem for a collection of 3D scene-understanding tasks using the Taskonomy dataset from Zamir et al. (2018). In the following, we will

[1]Test data and code can be found at https://goo.gl/uoXj8m

introduce the experiment setting and explain how we adapt the transferability metric to pixel-to-pixel recognition tasks. Then we compare transferability with *task affinity*, an empirical transferability metric proposed by Zamir et al. (2018).

**Data and Tasks.** The Taskonomy dataset contains 4,000,000 images of indoor scenes of 600 buildings. Every image has annotations for 26 computer vision tasks. We randomly sampled 20,000 images as training data. Eight tasks were chosen for this experiment, covering both classifications and lower-level pixel-to-pixel tasks. Table 6 summaries the specifications of these tasks and sample outputs are shown in Figure 5.

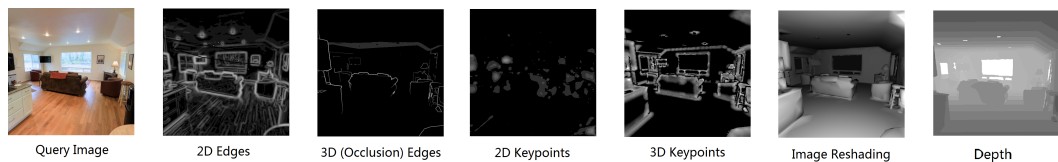

Figure 5: Ground truth of different tasks for a given query image.

**Feature Extraction and Data Preprocessing.** For each task, Zamir et al. (2018) trained a fully supervised network with an encoder-decoder structure. When testing the transfer performance from $\mathcal{T}_S$ to $\mathcal{T}_T$, the encoder output of $\mathcal{T}_S$ is used for training the decoder of $\mathcal{T}_T$. For a fair comparison, we use the same trained encoders to extract source features. The output dimension of all encoders are $16 \times 16 \times 8$ and we flatten the output into a vector of length 2048. To reduce the computational complexity, we also resize the ground truth images into $64 \times 64$.

For classification tasks, H-score can be easily calculated given the source features. But for pixel-to-pixel tasks such as Edges and Depth, their ground truths are represented as images, which can not be quantized easily. As a workaround, we cluster the pixel values in the ground truth images into a palette of 16 colors. Then compute the H-score of the source features with respect to each pixel, before aggregating them into a final score by averaging over the whole image.

We ran the experiment on a workstation with $3.40$ GHz $\times 8$ CPU and 16 GB memory. Each pairwise H-score computation finished in less than one hour including preprocessing. Then we rank the source tasks according to their H-scores of a given target task and compare the ranking with that in Zamir et al. (2018).

| Tasks | $k$ | Output | Quantize-level |
|---|---|---|---|
| 2D Edges | 2048 | images | 16 |
| 3D Occlusion Edges | 2048 | images | 16 |
| 2D Keypoints | 2048 | images | 16 |
| 3D Keypoints | 2048 | images | 16 |
| Reshading | 2048 | images | 16 |
| Depth | 2048 | images | 16 |
| Object Class. | 2048 | labels | none |
| Scene Class. | 2048 | labels | none |

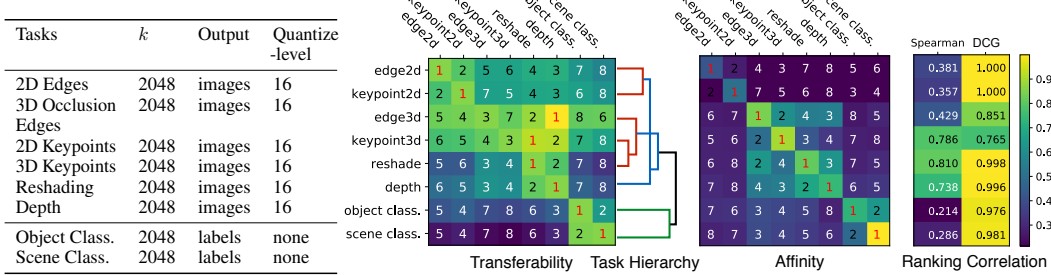

Figure 6: Task descriptions

Figure 7: Ranking comparison between transferability and affinity score.

**Pairwise Transfer Results.** Source task ranking results using transferability and affinity are visualized side by side in Figure 7, with columns representing source tasks and rows representing target tasks. For classification tasks (the bottom two rows in the transferability matrix), the top two transferable source tasks are identical for both methods. The best source task is the target task itself, as the encoder is trained on a task-specific network with much larger sample size. Scene Class. and Object Class. are ranked second for each other, as they are semantically related. Similar observations can be found in 2D pixel-to-pixel tasks (top two rows). The results on lower rankings are noisier.

A slightly larger difference between the two rankings can be found in 3D pixel-to-pixel tasks, especially 3D Occlusion Edges and 3D Keypoints. Though the top four ranked tasks of both methods

are exactly the four 3D tasks. It could indicate that these low level vision tasks are closely related to each other so that the transferability among them are inherently ambiguous. We also computed the ranking correlations between transferability and affinity using Spearman's R and Discounted Cumulative Gain (DCG). Both criterion show positive correlations for all target tasks. The correlation is especially strong with DCG as higher ranking entities are given larger weights.

The above observations inspire us to define a notion of task relatedness, as some tasks are frequently ranked high for each other. Specifically, we represent each task with a vector consisting of H-scores of all the source tasks, then apply agglomerative clustering over the task vectors. As shown in the dendrogram in Figure 7, 2D tasks and most 3D tasks are grouped into different clusters, but on a higher level, all pixel-to-pixel tasks are considered one category compared to the classifications tasks.

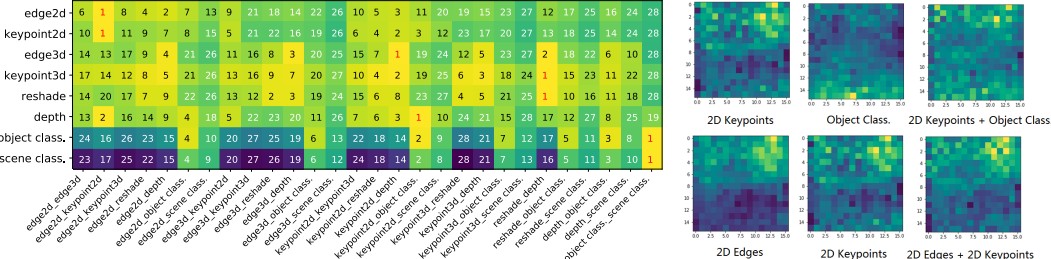

Figure 8: Ranking of second order transferability for all recognition tasks

Figure 9: First and second order pixel-wise transferability to Depth.

**Higher Order Transfer.** Sometimes we need to combine features from two or more source tasks for better transfer performance. A common way to combine features from multiple models in deep neural networks is feature concatenation. For such problems, our transferability definition can be easily adapted to high order feature transfer, by computing the H-score of the concatenated features.

Figure 8 shows the ranking results of all combinations of source task pairs for each target task. For all tasks except for 3D Occlusion Edges and Depth, the best seond-order source feature is the combination of the top two tasks of the first-order ranking. We examine the exception in Figure 9, by visualizing the pixel-by-pixel H-scores of first and second order transfers to Depth using a heatmap (lighter color implies a higher H-score). Note that different source tasks can be good at predicting different parts of the image. The top row shows the results of combining tasks with two different "transferability patterns" while the bottom row shows those with similar patterns. Combining tasks with different transferability patterns has a more significant improvement to the overall performance of the target task.

## 6 RELATED WORKS

**Transfer learning.** Transfer learning can be devided into two categories: *domain adaptation*, where knowledge transfer is achieved by making representations learned from one input domain work on a different input domain, e.g. adapt models for RGB images to infrared images (Wang & Deng (2018)); and *task transfer learning*, where knowledge is transferred between different tasks on the same input domain (Torrey & Shavlik (2010)). Our paper focus on the latter prolem.

**Empirical studies on transferability.** Yosinski et al. (2014) compared the transfer accuracy of features from different layers in a neural network between image classification tasks. A similar study was performed for NLP tasks by Conneau et al. (2017). Zamir et al. (2018) determined the optimal transfer hierarchy over a collection of perceptual indoor scene understanidng tasks, while transferability was measured by a non-parameteric score called "task affinity" derived from neural network transfer losses coupled with an ordinal normalization scheme.

**Task relatedness.** One approach to define task relatedness is based on task generation. Generalization bounds have been derived for multi-task learning (Baxter (2000)), learning-to-learn (Maurer (2009)) and life-long learning (Pentina & Lampert (2014)). Although these studies show theoretical results on transferability, it is hard to infer from data whether the assumptions are satisfied. Another approach is estimating task relatedness from data, either explicitly (Bonilla et al. (2008); Zhang

(2013)) or implicitly as a regularization term on the network weights (Xue et al. (2007); Jacob et al. (2009)). Most works in this category are limited to shallow ones in terms of the model parameters.

**Representation learning and evaluation.** Selecting optimal features for a given task is traditionally performed via feature subset selection or feature weight learning. Subset selection chooses features with maximal relevance and minimal redundancy according to information theoretic or statistical criteria (Peng et al. (2005); Hall (1999)). The feature weight approach learns the task while regularizing feature weights with sparsity constraints, which is common in multi-task learningLiao & Carin (2006); Argyriou et al. (2007). In a different perspective, Huang et al. (2017) consider the universal feature selection problem, which finds the most informative features from data when the exact inference problem is unknown. When the target task is given, the universal feature is equivalent to the minimum error probability feature used in this work.

## 7   CONCLUSION

In this paper, we presented H-score, an information theoretic approach to estimating the performance of features when transferred across classification tasks. Then we used it to define a notion of task transferability in multi-task transfer learning problems, that is both time and sample complexity efficient. The resulting transferability metric also has a strong operational meaning as the ratio between the best achievable error exponent of the transferred representation and the minium error exponent of the target task.

Our transferability score successfully predicted the performance for transfering features from ImageNet-1000 classification task to Cifar-100 task. Moreover, we showed how the transferability metric can be applied to a set of diverse computer vision tasks using the Taskonomy dataset.

In future works, we plan to extend our theoretical results to non-classification tasks, as well as relaxing the local assumptions on the conditional distributions of the tasks. We will also investigate properties of higher order transferability, developing more scalable algorithms that avoid computing the H-score of all task pairs. On the application side, as transferability tells us how different tasks are related, we hope to use this information to design better task hierarchies for transfer learning.

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

## APPENDIX A    INFERENCE USING LOCAL INFORMATION GEOMETRY

### A.1    A PRIMER ON ERROR EXPONENTS

To begin with, consider the binary hypothesis testing problem over $m$ i.i.d. sampled observations $\{x^{(i)}\}_{i=1}^m \triangleq x^m$ with the following hypotheses: $H_0 : x^m \sim P_1$ or $H_1 : x^m \sim P_2$. Let $P_{x^m}$ be the empirical distribution of the samples. The optimal test, i.e., the log likelihood ratio test can be stated in terms of information-theoretic quantities as follows:

$$\log \frac{P_1(x^m)}{P_2(x^m)} = m[D(P_{x^m}||P_2) - D(P_{x^m}||P_1)] \underset{H_1}{\overset{H_0}{\gtrless}} \log T$$

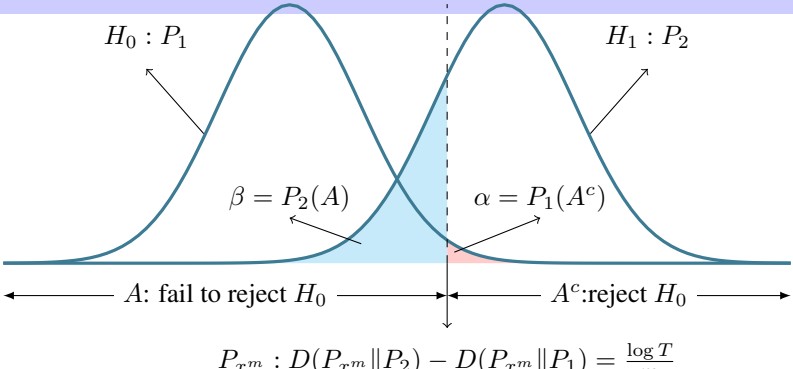

$$P_{x^m} : D(P_{x^m}||P_2) - D(P_{x^m}||P_1) = \frac{\log T}{m}$$

Figure 10: The binary hypothesis testing problem. The blue curves shows the probility density functions for $P_1$ and $P_2$. The rejection region and the acceptance region are highlighted in red and blue, respectively. The vertical line indicates the decision threshold.

Further, using Sannov's theorem, we have that asymptotically the probability of type I error

$$\alpha = P_1(A^c) \approx 2^{-mD(P_1^*||P_1)}$$

where $P_1^* = \text{argmin}_{P \in A^c} D(P||P_1)$ and $A^c(T) = \{x^m : D(P_{x^m}||P_2) - D(P_{x^m}||P_1) < \frac{1}{m}\log T\}$ denotes the rejection region. Similarly, for type II error

$$\beta = P_2(A) \approx 2^{-mD(P_2^*||P_2)}$$

where $P_2^* = \text{argmin}_{P \in A} D(P||P_2)$ and $A = \{x^m : D(P_{x^m}||P_2) - D(P_{x^m}||P_1) > \frac{1}{m}\log T\}$ represents the acceptance region. (See Figure 10) The overall probability of error is $P_e^{(m)} = \alpha Pr(H_0) + \beta Pr(H_1)$ and the *best achievable exponent in the Bayesian probability of error (a.k.a. Chernoff exponent)* is defined as:

$$E = \lim_{m \to \infty} \min_{A \subseteq \mathcal{X}^m} -\frac{1}{m} \log P_e^{(m)}$$

See Cover & Thomas (1991) for more background information on error exponents and its related theorems.

## A.2 Local information geometry

Now consider the same binary hypothesis testing problem, but with the local constraint $P_1, P_2 \in \mathcal{N}_\epsilon(P_0^{\mathcal{X}})$. Let $\phi_i = \frac{P_i(x) - P_0^{\mathcal{X}}(x)}{\epsilon \sqrt{P_0^{\mathcal{X}}(x)}}$ denote the information vectors corresponding to $P_i$ for $i = 1, 2$.

Makur et al. (2015) uses local information geometry to connect the error exponent in hypothesis testing to the length of certain information vectors, summarized in the following two lemmas.

**Lemma 1.** *Given zero-mean, unit variance feature function $f(x) : \mathcal{X} \to \mathbb{R}$, the optimal error exponent (a.k.a. Chernoff exponent) of this hypothesis testing problem is*

$$E = \frac{\epsilon^2}{8} \|\phi_1 - \phi_2\|^2 + o(\epsilon^2)$$

**Lemma 2.** *Given zero-mean, unit variance feature function $f(x) : \mathcal{X} \to \mathbb{R}$, the error exponent of a mismatched decision function of the form $l = \frac{1}{m} \sum_{i=1}^m (f(x^{(i)}))$ is*

$$E_f = \frac{\epsilon^2}{8} \langle \xi, \phi_1 - \phi_2 \rangle^2 + o(\epsilon^2)$$

*where $\xi(x) = \sqrt{P_0(x)} f(x)$ is the feature vector associated with $f(x)$.*

As our discussion of transferability mostly concerns with multi-dimensional features, we present the $k$-dimensional generalization of Lemma 2 below: (Equation 1 in the main paper.)

**Lemma 3.** *Given $k$ normalized feature functions $f(x) = [f_1(x), \dots, f_k(x)]$, such that $\mathbb{E}[f_i(X)] = 0$ for all $i$, and $\text{cov}(f(X)) = I$, we define a $k$-d statistics of the form $l^k = (l_1, \dots, l_k)$ where $l_i = \frac{1}{m} \sum_{l=1}^m f_i(x^{(l)})$. Let $\Xi(x) = [\xi_1(x), \dots, \xi_k(x)]$ be the corresponding feature vectors with $\xi_i(x) = \sqrt{P_X(x)} f_i(x)$.*

$$E_f^k = \sum_{i=1}^k E_{f_i} = \sum_{i=l}^k \frac{\epsilon^2}{8} \langle \xi_i, \phi_1 - \phi_2 \rangle^2 + o(\epsilon^2) \tag{4}$$

*Proof.* According to Cramér's theorem, the error exponent under $P_i$ is

$$E_i(\lambda) = \min_{P \in \delta(\lambda)} D(P \| P_i)$$

where $\delta(\lambda) \triangleq \{P : \mathbb{E}_P[f(X)^k] = \lambda \mathbb{E}_{P_1}[f(X)^k] + (1 - \lambda) \mathbb{E}_{P_2}[f(X)^k]\}$. With the techniques developed in local information geometry, the above above problem is equivalent to the following problem:

$$E_i(\lambda) = \min_{\tilde{\phi}_{\theta^k} \in \sigma(\lambda)} \frac{1}{2} \|\tilde{\phi}_{\theta^k} - \phi_i\|^2$$

where $\sigma(\lambda) \triangleq \{\tilde{\phi}_{\theta^k} : \langle \tilde{\phi}_{\theta^k}, \xi_l \rangle = \langle \lambda \phi_1 + (1 - \lambda) \phi_2, \xi_l \rangle, l = 1, \dots, k\}$ and $i = 1, 2$. Further, using the equation $\tilde{\phi}_{\theta^k}(x) = \sum_{l=1}^k \theta_l \xi_l(x) + \phi_i(x) - \alpha(\theta^k) \sqrt{P_0(x)} + o(\epsilon)$ and the equation $\alpha(\theta^k) = o(\epsilon^2)$, it is easy to show that $E_1(\lambda) = E_1(\lambda)$ when $\lambda = \frac{1}{2}$. Then the overall error probability has the exponent as shown in Equation (1). □

# Appendix B   HGR Maximal Correlation and The ACE Algorithm

Given random variables $X$ and $Y$, the HGR maximal correlation $\rho(X; Y)$ defined in Equation 2 is a generalization of the Pearson's correlation coefficient to capture non-linear dependence between random variables. According to Rényi (1959), it satisfies all seven natural postulates of a suitable dependence measure. Some notable properties are listed below:

1. $\rho(X;Y)$ is defined for any pair of random variables $X$ and $Y$
2. $0 \le \rho(X;Y) \le 1$;
3. $\rho(X;Y) = 0$ if and only if $X$ and $Y$ are independent
4. $\rho(X;Y) = 1$ if and only if $X$ and $Y$ are strictly dependent.

When the feature dimension is 1, the solution of the maximal HGR correlation is $\rho(X;Y) = \sigma_1$, the largest singular value of the DTM matrix $\tilde{B}$. For $k$-dimensional features, $\rho(X;Y) = \sum_i^k (\sigma_i)$. However, computing $\tilde{B}$ requires estimating the joint probability $P_{YX}$ from data, which is inpractical in real applications. Breiman & Friedman (1985) proposed an efficient algorithm, alternating condition expectation (ACE), inspired by the power method for computing matrix singular values.

---

**Algorithm 1** The ACE algorithm

---

**Require:** training samples $\{((x^{(i)}, y^{(i)}) : i = 1, \ldots, m\}$
 1: Initialize $g(y) \in \mathbb{R}^k, \forall y \in \mathcal{Y}$ randomly
 2: Center $g(y) \leftarrow g(y) - \mathbb{E}[g(y)]$
 3: **repeat**
 4:     $f(x) \leftarrow \mathbb{E}[g(Y) \mid X = x], \forall x \in \mathcal{X}$
 5:     $f(x) \leftarrow f(x)\mathbb{E}[f(X)f(X)^T]^{-\frac{1}{2}}, \forall x \in \mathcal{X}$ {Normalize $f(x)$}
 6:     $g(y) \leftarrow \mathbb{E}[f(X) \mid Y = y], \forall y \in \mathcal{Y}$
 7:     $g(y) \leftarrow g(x)\mathbb{E}[g(Y)g(Y)^T]^{-\frac{1}{2}}, \forall y \in \mathcal{Y}$ {Normalize $g(y)$}
 8: **until** $\mathbb{E}[f(X)^T g(Y)]$ stops increasing

---

In Algorithm 1, we first initialize $g$ as a random $k$-dimensional zero-mean function. Then iteratively update $f(x)$ and $g(y)$ for all $x \in X$ and $y \in Y$. The conditional distributions on Line 4 and 6 are computed as the empirical average over $m$ samples. The normalization steps on Lines 5 and 7 can also be implemented using the Gram-Schmidt process. Note that the ACE algorithm has several variations in previous works, including a kernel-smoothed version (Breiman & Friedman (1985)) and a parallel version with improved efficiency (Huang et al. (2017)). An alternative formulation that supports continuous $X$ and large feature dimension $k$ has also been proposed recently (Wang et al. (2019)).

Next we look at the convergence property of the ACE algorithm. Let $f(X), g(Y)$ be the true maximal correlation functions, and let $\tilde{f}(X), \tilde{g}(Y)$ be estimations computed with Algorithm 1 from $m$ i.i.d. sampled training data. Similarly, denote by $\mathbb{E}[f(X)g(Y)]$ and $\hat{\mathbb{E}}_m[\tilde{f}(X)\tilde{g}(Y)]$ the true and estimated maximal correlations, respectively. Using Sanov's Theorem, we can show that for a small $\Delta > 0$, the probability that the ratio between the true and estimated maximal correlation is within $1 \pm \Delta$ drops exponentially as the number of samples increases. Hence the ACE algorithm converges in exponential time. The following theorem gives the precise sampling complexity for $k = 1$.

**Theorem 2.** *For any random variables $X$ and $Y$ with joint probability distribution $P_{YX}$, if $X$ and $Y$ are not independent, then for any $f : \mathcal{X} \to \mathbb{R}$ and $g : \mathcal{Y} \to \mathbb{R}$ such that $\mathbb{E}[f^2(X)] = \mathbb{E}[g^2(Y)] = 1$, we have that*

$$-\lim_{\Delta \to 0^+} \frac{1}{\delta^2} \lim_{m \to \infty} \frac{1}{m} \log \left[ P \left\{ \left| \frac{\hat{\mathbb{E}}_m[\tilde{f}(X)\tilde{g}(Y)]}{\mathbb{E}[f(X)g(Y)]} - 1 \right| \ge \Delta \right\} \right] = \frac{1}{2} \frac{\mathbb{E}[f(X)g(Y)]^2}{var[f(X)g(Y)]}$$

*for any given $\Delta > 0$.*

## APPENDIX C    PROOF OF THEOREM 1

To simplify the proof, we first consider the case when the feature function is 1-dimensional.i.e. $f : \mathcal{X} \to \mathbb{R}$. We have the following lemma:

**Lemma 4.** *Let $f : X \rightarrow \mathbb{R}$ be an arbitrary feature function of $X$ such that $\mathbb{E}[f(X)] = 0$ and $\mathbb{E}[f(X)^2] = 1$, then $||\tilde{B}\xi||_F^2 = \mathbb{E}_{P_Y}[(\mathbb{E}[f(X)|Y])^2]$ where $\xi$ is the feature vector corresponding to $f$.*

*Proof.* Since $\xi(x) = \sqrt{P_X(x)}f(x)$, we have

$$||\tilde{B}\xi||_F^2 = \xi^{\mathrm{T}}\tilde{B}^{\mathrm{T}}\tilde{B}\xi$$

$$= \sum_{y\in\mathcal{Y}}\left(\sum_{x\in\mathcal{X}}\left(\frac{P_{YX}(y,x)}{\sqrt{P_X(x)}\sqrt{P_Y(y)}} - \sqrt{P_X(x)}\sqrt{P_Y(y)}\right) \cdot \sqrt{P_X(x)}f(x)\right)^2$$

$$= \sum_{y\in\mathcal{Y}}\left(\sum_{x\in\mathcal{X}}\frac{P_{YX}(y,x)}{P_Y(y)} \cdot f(x) - \sum_{x\in\mathcal{X}}f(x)P_X(x)\right)^2 \cdot P_Y(y)$$

$$= \sum_{y\in\mathcal{Y}}(\mathbb{E}[f(X)|Y=y] - \mathbb{E}[f(x)])^2 \cdot P_Y(y)$$

$$= \mathbb{E}_{P_Y}[(\mathbb{E}[f(X)|Y])^2]$$

The last equality uses the assumption that $\mathbb{E}[f(x)] = 0$. $\qquad\square$

**Theorem 3** (1D version of Theorem 1). *Given $P_{X|Y=0}, P_{X|Y=1} \in \mathcal{N}_\epsilon^{\mathcal{X}}(P_{0,X})$ and features $f : X \rightarrow \mathbb{R}$ such that $\mathbb{E}[f(X)] = 0$ and $\mathbb{E}[f(X)^2] = 1$, then there exists some constant $c$ independent of $f$ such that*

$$E_f = c||\tilde{B}\xi||_F^2 \tag{5}$$

*Proof.* From $\mathbb{E}[f(X)] = P_Y(0)\mathbb{E}[f(X)|Y=0] + P_Y(1)\mathbb{E}[f(X)|Y=1] = 0$, we first derive the following properties of the conditional expectations of $f(x)$:

$$\mathbb{E}[f(X)|Y=1] = -\left(\frac{P_Y(0)}{P_Y(1)}\right)\mathbb{E}[f(X)|Y=0]$$

$$\mathbb{E}[f(X)|Y=0] = -\left(\frac{P_Y(1)}{P_Y(0)}\right)\mathbb{E}[f(X)|Y=1]$$

On the R.H.S. of Equation 5, we apply Lemma 4 to write

$$||\tilde{B}\xi||_F^2 = \mathbb{E}_{P_Y}\left[(\mathbb{E}[f(X)|Y])^2\right]$$

$$= \frac{P_Y(0)P_Y(1) + P_Y(1)^2}{P_Y(0)}(\mathbb{E}[f(X)|Y=1])^2$$

Next consider the L.H.S. of the equation, by Lemma 2, we have $E(h) = \frac{\epsilon^2}{8}\langle\xi, \phi_1 - \phi_2\rangle^2 + o(\epsilon^2) = c_0\langle\xi, \phi_1 - \phi_2\rangle^2$ for some constant $c_0$.

$$c_0\langle\xi, \phi_{P_1} - \phi_{P_2}\rangle^2$$

$$= c_0\left(\sum_{x\in\mathcal{X}}\frac{P_{X|Y=0}(x) - P_{X|Y=1}(x)}{\sqrt{P_X(x)}} \cdot \sqrt{P_X(x)}f(x)\right)^2$$

$$= c_0\left(\mathbb{E}[f(X)|Y=0] - \mathbb{E}[f(X)|Y=1]\right)^2$$

$$= c_0\frac{-(P_Y(0) + P_Y(1))^2}{P_Y(0)P_Y(1)} \cdot (\mathbb{E}[f(X)|Y=1])(\mathbb{E}[f(X)|Y=0])$$

$$= c_0\left(\frac{P_Y(0) + P_Y(1)}{P_Y(0)}\right)^2 \cdot (\mathbb{E}[f(X)|Y=1])^2$$

$$= c_0\frac{P_Y(0) + P_Y(1)}{P_Y(0)P_Y(1)}\left(\frac{P_Y(0)P_Y(1) + P_Y(1)^2}{P_Y(0)}\right) \cdot (\mathbb{E}[f(X)|Y=1])^2$$

$$= c\,||\tilde{B}\xi||_F^2$$

$\qquad\square$

For $k \geq 2$, Lemma 4 can be restated as follows:

**Lemma 5.** *Let $f : X \rightarrow \mathbb{R}^k$ be a $k$-dimensional feature function of $X$, where $f(x) = [f_1(x), \ldots, f_k(x)]$. Further we assume that $\mathbb{E}[f(X)] = 0$ and $\mathrm{cov}(f(X)) = \mathbb{E}[f(X)^{\mathrm{T}} f(X)] = I$. Then $||\hat{B}\Xi||_F^2 = \mathrm{tr}(\mathrm{cov}(\mathbb{E}[f(X)|Y]))$ where columns of $\Xi$ are the information vectors corresponding to $f_1(x), \ldots, f_k(x)$.*

*Proof.* First, note that

$$
\tilde{B}\Xi = \left( \left[ \sqrt{\mathrm{P}_Y} \right]^{-1} \mathrm{P}_{YX} \left[ \sqrt{\mathrm{P}_X} \right]^{-1} - \sqrt{\mathrm{P}_Y} \sqrt{\mathrm{P}_X}^{\mathrm{T}} \right) \left[ \sqrt{\mathrm{P}_X} \right] \mathrm{f}(X)
$$
$$
= \left[ \sqrt{\mathrm{P}_Y} \right] \left( [\mathrm{P}_Y]^{-1} \mathrm{P}_{YX} \mathrm{f}(X) - \mathbf{1} \cdot \mathbb{E}[f(X)]^{\mathrm{T}} \right)
$$

where $\mathbf{1}$ is a column vector with all entries 1 and length $|\mathcal{Y}|$. Since $\mathbb{E}[f(X)] = 0$, we have

$$
\tilde{B}\Xi = \left[ \sqrt{\mathrm{P}_Y} \right] \left( [\mathrm{P}_Y]^{-1} \mathrm{P}_{YX} \mathrm{f}(X) \right)
$$

It follows that

$$
\begin{aligned}
||\tilde{B}\Xi||_F^2 &= \mathrm{tr}(\Xi^{\mathrm{T}} \tilde{B}^{\mathrm{T}} \tilde{B}\Xi) \\
&= \mathrm{tr}\left( \left( [\mathrm{P}_Y]^{-1} \mathrm{P}_{YX} \mathrm{f}(X) \right)^{\mathrm{T}} [\mathrm{P}_Y] \left( [\mathrm{P}_Y]^{-1} \mathrm{P}_{YX} \mathrm{f}(X) \right) \right) \\
&= \mathrm{tr}\left( \mathbb{E}_{P_Y} \left[ (\mathbb{E}[f(X)|Y]]^{\mathrm{T}})^{\mathrm{T}} (\mathbb{E}[f(X)|Y]]^{\mathrm{T}}) \right] \right) \\
&= \mathrm{tr}\left( \mathrm{cov}(\mathbb{E}[f(X)|Y]) \right)
\end{aligned}
$$

$\square$

Finally, we derive the multi-dimensional case for Theorem 1.

*Proof of Theorem 1.* Using Lemma 5 and a similar argument as in the simplified proof, the R.H.S of the equation becomes

$$
\begin{aligned}
||\tilde{B}\Xi||_F^2 &= \mathrm{tr}\left( \mathrm{cov}(\mathbb{E}[f(X)|Y]) \right) \\
&= \mathrm{tr}\left( \mathbb{E}_{P_Y} \left[ (\mathbb{E}[f(X)|Y]]^{\mathrm{T}})^{\mathrm{T}} (\mathbb{E}[f(X)|Y]]^{\mathrm{T}}) \right] \right) \\
&= P_Y(0)\mathbb{E}[f(X)|Y = 0]]^{\mathrm{T}}\mathbb{E}[f(X)|Y = 0]] + P_Y(1)\mathbb{E}[f(X)|Y = 1]]^{\mathrm{T}}\mathbb{E}[f(X)|Y = 1]] \\
&= \frac{P_Y(0)P_Y(1) + P_Y(1)^2}{P_Y(0)} (\mathbb{E}[f(X)|Y = 1])^{\mathrm{T}} (\mathbb{E}[f(X)|Y = 1])
\end{aligned}
$$

By Lemma 3, the L.H.S. of the equation can be written as $E_f^k = c_0 \sum_{i=l}^{k} \langle \xi_i, \phi_1 - \phi_2 \rangle^2$ for some constant $c_0$. It follows that

$$
\begin{aligned}
& c_0 \sum_{i=l}^{k} \langle \xi_i, \phi_1 - \phi_2 \rangle^2 \\
&= c_0 \left( \left( (\mathrm{P}_{X|Y=0} - \mathrm{P}_{X|Y=1}) \right)^{\mathrm{T}} \mathrm{f}(X) \right) \left( \left( (\mathrm{P}_{X|Y=0} - \mathrm{P}_{X|Y=1}) \right)^{\mathrm{T}} \mathrm{f}(X) \right)^{\mathrm{T}} \\
&= c_0 \left( \mathbb{E}[f(X)|Y = 0] - \mathbb{E}[f(X)|Y = 1] \right)^{\mathrm{T}} \left( \mathbb{E}[f(X)|Y = 0] - \mathbb{E}[f(X)|Y = 1] \right) \\
&= c_0 \frac{P_Y(0) + P_Y(1)}{P_Y(0)P_Y(1)} \left( \frac{P_Y(0)P_Y(1) + P_Y(1)^2}{P_Y(0)} \right) \cdot \mathbb{E}[f(X)|Y = 1]^{\mathrm{T}} \mathbb{E}[f(X)|Y = 1] \\
&= c \, ||\tilde{B}\Xi||_F^2
\end{aligned}
$$

$\square$

## APPENDIX D    GENERALIZATION OF H-SCORE FOR ARBITRARY FEATURE SET

Since $\Xi = \left[\sqrt{P_X}\right] f(X)$ and $\mathbb{E}[f(X)] = 0$, we have

$$\Xi^{\mathrm{T}}\Xi = \left(\left[\sqrt{P_X}\right] f(X)\right)^{\mathrm{T}}\left(\left[\sqrt{P_X}\right] f(X)\right) = \mathbb{E}[f(X)^{\mathrm{T}}f(X)] = \mathrm{cov}(f(X)) \tag{6}$$

Equation (6) gives a more understandable expression of the normalization term. We can also write $\tilde{B}\Xi$ as follows:

$$\tilde{B}\Xi = \left(\left[\sqrt{P_Y}\right]^{-1} P_{YX}\left[\sqrt{P_X}\right]^{-1} - \sqrt{P_Y}\sqrt{P_X}^{\mathrm{T}}\right)\left[\sqrt{P_X}\right] f(X)$$

$$= \left[\sqrt{P_Y}\right]\left([P_Y]^{-1} P_{YX}f(X) - \mathbf{1} \cdot \mathbb{E}[f(X)]^{\mathrm{T}}\right)$$

where $\mathbf{1}$ is a column vector with all entries $1$ and length $|\mathcal{Y}|$, we have

$$\Xi^{\mathrm{T}}\tilde{B}^{\mathrm{T}}\tilde{B}\Xi = \left([P_Y]^{-1} P_{YX}f(X) - \mathbf{1} \cdot \mathbb{E}[f(X)]^{\mathrm{T}}\right)^{\mathrm{T}}[P_Y]\left([P_Y]^{-1} P_{YX}f(X) - \mathbf{1} \cdot \mathbb{E}[f(X)]^{\mathrm{T}}\right)$$

$$= \mathbb{E}_{P_Y}\left[(\mathbb{E}[f(X)|Y] - \mathbf{1} \cdot \mathbb{E}[f(X)]^{\mathrm{T}})^{\mathrm{T}}(\mathbb{E}[f(X)|Y] - \mathbf{1} \cdot \mathbb{E}[f(X)]^{\mathrm{T}})\right]$$

$$= \mathrm{cov}\left(\mathbb{E}[f(X)|Y]\right) \tag{7}$$

On the other hand,

$$\|\tilde{B}\Xi(\Xi^{\mathrm{T}}\Xi)^{-\frac{1}{2}}\|_F^2 = \mathrm{tr}\left((\Xi^{\mathrm{T}}\Xi)^{-\frac{1}{2}}\Xi^{\mathrm{T}}\tilde{B}^{\mathrm{T}}\tilde{B}\Xi(\Xi^{\mathrm{T}}\Xi)^{-\frac{1}{2}}\right)$$

$$= \mathrm{tr}\left((\Xi^{\mathrm{T}}\Xi)^{-1}\Xi^{\mathrm{T}}\tilde{B}^{\mathrm{T}}\tilde{B}\Xi\right) \tag{8}$$

$$= \mathrm{tr}\left(\mathrm{cov}(f(X))^{-1}\mathrm{cov}(\mathbb{E}[f(X)|Y])\right)$$

The last equality is derived by substituting (6) and (7) into (8).

## APPENDIX E    H-SCORE AND SOFTMAX REGRESSION

In softmax regression, given $m$ training examples $\{(x^{(i)}, y^{(i)})\}_{i=1}^m$, the cross-entropy loss of the model is

$$\ell(f, \theta) = -\sum_{i=1}^m \sum_{k=1}^{|\mathcal{Y}|} \mathbf{1}\{y^{(i)} = k\} \log \frac{e^{-\theta_k^T f(x^{(i)})}}{\sum_{j=1}^C e^{-\theta_j^T f(x^{(i)})}} \tag{9}$$

$$\triangleq -\sum_{i=1}^m \log Q_{Y|X}(y^{(i)}|x^{(i)}) \tag{10}$$

$$= -\mathbb{E}_{P_{YX}}[\log Q_{Y|X}(Y|X)] \tag{11}$$

Minimizing $\ell$ is equivalent to minimizing $D\left(P_{YX}\|P_X Q_{Y|X}\right)$ where $P_{YX}$ is the joint empirical distribution of $(X, Y)$.

Using information geometry, it can be shown that under a local assumption

$$\underset{f,\theta}{\mathrm{argmin}}\, D\left(P_{YX}\|P_X Q_{Y|X}\right) = \underset{\Psi,\Phi}{\mathrm{argmin}}\, \frac{1}{2}\|\tilde{B} - \Psi\Phi^{\mathrm{T}}\|_F^2 + o(\epsilon^2) \tag{12}$$

which reveals a close connection between log loss and the modal decomposition of $\tilde{B}$. In consequence, it is reasonable to measure the classification performance with $\|\tilde{B} - \Psi\Phi^{\mathrm{T}}\|_F^2$ given a pair of $(f, \theta)$ associated with $(\Psi, \Phi)$.

In the context of estimating transferability, we are interested in a one-sided problem, where $\Phi_S$ is given by the source feature and $\Psi$ becomes the only variable.

$$\min_{\Psi} \|\tilde{B}_T - \Psi\Phi_S^{\mathrm{T}}\| \tag{13}$$

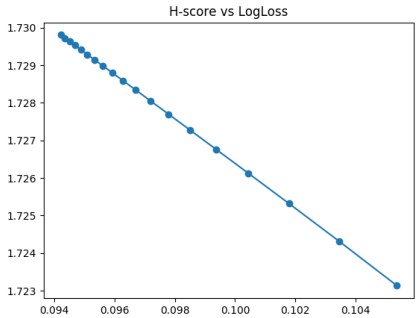

Figure 11: Comparing task transferability (H-score) and transfer log-loss on synthesized tasks.

Training the network is equivalent to finding the optimal weight $\Psi^*$ that minimizes the log-loss. By taking the derivative of the Objective function with respect to $\Psi$, we get

$$\Psi^* = \tilde{B}_T \Phi_S (\Phi_S^T \Phi_S)^{-1} \tag{14}$$

Substituting (14) in the Objective of (13), we can derive the following close-form solution for the log loss.

$$\|\tilde{B}_T^T \tilde{B}_T\|_F^2 - \|\tilde{B}_T \Phi_S (\Phi_S^T \Phi_S)^{-\frac{1}{2}}\|_F^2 \tag{15}$$

The first term in (15) is fixed given $\mathcal{T}_T$ while the second term has exactly the form of H-score. This implies that log loss is negatively linearly related to H-score.

We demonstrates this relationship experimentally, using a collection of synthesized tasks (Figure 11). In particular, the target task is generated based on a random stochastic matrix $P_{Y_0|X}$, and 20 source tasks are generated with the conditional probability matrix $P_{Y_i|X} = P_{Y_0|X} + i\lambda I$ for some positive constant $\lambda$.

The universal minimum error probability features for each source task are used as the source features $f_{S_i}(x)$, while the respective log-loss are obtained through training a simple neural network in Figure 2 with cross-entropy loss. The relationship is clearly linear with a constant offset.

## APPENDIX F  DTM AND MUTUAL INFORMATION

**Proposition 1.** *Under the local assumption that $P_{X|Y} \in N_\epsilon(P_X)$, mutual information $I(X;Y) = \frac{1}{2}\|\tilde{B}\|_F^2 + o(\epsilon^2)$, where $\tilde{B}$ of the DTM matrix of $X$ and $Y$.*

*Proof.* First, we define $\phi_y^{X|Y}(x) = \frac{P_{X|Y}(x|y) - P_X(x)}{\epsilon\sqrt{P_X(x)}}$ and let $\Phi^{X|Y} \in \mathbb{R}^{|\mathcal{X}| \times |\mathcal{Y}|}$ denote its matrix version. Then we have

$$\epsilon\Phi^{X|Y}[\sqrt{P_Y}] = [\sqrt{P_X}]^{-1}(P_{XY} - P_X)[\sqrt{P_Y}] = \tilde{B}^T$$

Next, we express the mutual information in terms of information vector $\phi^{X|Y}$,

$$I(X;Y) = I(Y;X) = \sum_{y \in \mathcal{Y}} P_Y(y) D_{KL}(P_{X|Y}\|P_X)$$

$$= \frac{\epsilon^2}{2} \sum_{y \in \mathcal{Y}} P_Y(y)\|\phi_y^{X|Y}\|^2 + o(\epsilon^2)$$

$$= \frac{1}{2}\|\epsilon\Phi^{X|Y}[\sqrt{P_Y}]\|_F^2 + o(\epsilon^2)$$

$$= \frac{1}{2}\|\tilde{B}\|_F^2 + o(\epsilon^2)$$

$\square$

## APPENDIX G  SUPPLEMENTARY RESULTS ON EXPERIMENT 5.2

### G.1  COMPARISON OF H-SCORES AND AFFINITIES

Table 1, with columns representing source tasks and rows representing target tasks. For each target task, the upper row shows our results while the lower one shows the results in Zamir et al. (2018). Score values are included in parenteses.

Table 1: Transferability ranking comparison, between H-score's estimation and task affinity

| Tasks | 2D Edges | 2D Keypoints | 3D Edges | 3D Keypoints | Reshading | Depth | Object Class. | Scene Class. |
|---|---|---|---|---|---|---|---|---|
| 2D Edges | 1 (1.8216) | 2 (1.7334) | 5 (1.5704) | 6 (1.5696) | 4 (1.6146) | 3 (1.6201) | 7 (1.5097) | 8 (1.4402) |
| | 1 (0.0389) | 2 (0.0117) | 4 (5.8920e-5) | 3 (8.8011e-5) | 7 (2.9001e-5) | 8 (2.2110e-5) | 5 (4.9141e-5) | 6 (4.8720e-5) |
| 2D Keypoints | 2 (1.6698) | 1 (1.7859) | 7 (1.5248) | 5 (1.5287) | 4 (1.5481) | 3 (1.5632) | 6 (1.5253) | 8 (1.4725) |
| | 2 (0.0002) | 1 (0.0542) | 7 (7.7797e-5) | 5 (8.1029e-5) | 6 (7.8464e-5) | 8 (7.2724e-5) | 3 (0.0002) | 4 (0.0001) |
| 3D Edges | 5 (1.4828) | 4 (1.4910) | 3 (1.5167) | 7 (1.4701) | 2 (1.5405) | 1 (1.6739) | 8 (1.4644) | 6 (1.4730) |
| | 6 (0.0117) | 7 (0.0108) | 1 (0.1179) | 2 (0.0734) | 4 (0.0622) | 3 (0.0636) | 8 (0.0094) | 5 (0.0151) |
| 3D Keypoints | 6 (1.5375) | 5 (1.5466) | 4 (1.5910) | 3 (1.6456) | 1 (1.7198) | 2 (1.7122) | 7 (1.4709) | 8 (1.4121) |
| | 5 (0.0141) | 6 (0.0136) | 2 (0.0531) | 1(0.1275) | 3 (0.0400) | 4 (0.0247) | 7 (0.0132) | 8 (0.0121) |
| Reshading | 5 (1.5504) | 6 (1.5426) | 3 (1.8174) | 4 (1.7990) | 1 (2.2339) | 2 (2.1200) | 7 (1.4774) | 8 (1.3804) |
| | 6 (0.0147) | 8 (0.0143) | 2 (0.0781) | 4(0.0545) | 1 (0.1121) | 3 (0.0765) | 7 (0.0144) | 5 (0.0174) |
| Depth | 6 (1.6542) | 5 (1.6870) | 3 (1.8504) | 4 (1.8176) | 2 (2.1700) | 1 (2.2441) | 7 (1.6008) | 8 (1.5099) |
| | 7 (0.0175) | 8 (0.0154) | 3 (0.0595) | 4 (0.0617) | 2 (0.0867) | 1 (0.0989) | 6 (0.0217) | 5 (0.0237) |
| Object Class. | 5 (22.866) | 4 (23.627) | 7 (22.371) | 8 (21.950) | 6 (22.452) | 3 (23.697) | 1 (33.468) | 2 (28.013) |
| | 7 (0.0205) | 6 (0.0217) | 3 (0.0350) | 4 (0.0318) | 5 (0.0286) | 8 (0.0147) | 1 (0.0959) | 2 (0.0774) |
| Scene Class. | 5 (14.575) | 4 (15.074) | 7 (14.206) | 8 (13.801) | 6 (14.332) | 3 (15.474) | 2 (25.750) | 1 (25.962) |
| | 8 (0.0149) | 7 (0.0165) | 3 (0.0335) | 4 (0.0305) | 5 (0.0263) | 6 (0.0198) | 2 (0.0504) | 1 (0.1474) |

Here we present some detailed results on the comparison between H-score and the affinity score in Zamir et al. (2018) for pairwise transfer.

The results of the classification tasks are shown in Figure 12 and the results of Depth is shown in 13. We can see in general, although affinity and transferability have totally different value ranges, they tend to agree on the top few ranked tasks.

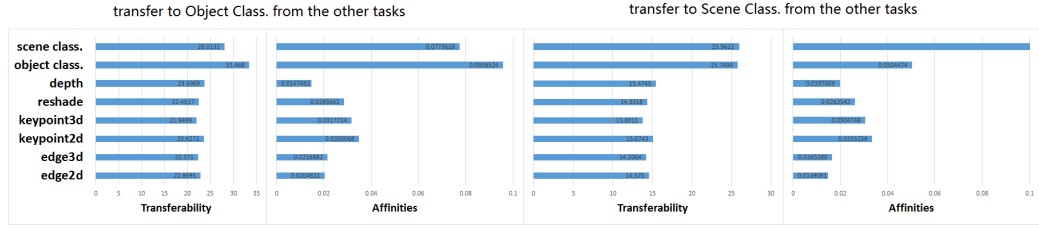

Figure 12: Source task transferability ranking for classification tasks. For each target task, the left figure shows H-score results, and the right figure shows task affinity results.

### G.2  LABEL QUANTIZATION

During the quantization process of the pixel-to-pixel task labels (ground truth images), we are primarily concerned with two factors: computational complexity and information loss. Too much information loss will lead to bad approximation of the original problems. On the other hand, having little information loss requires larger cluster size and computation cost.

Figure (14) shows that even after quantization, much of the information in the images are retained.

To test the sensitivity of the cluster size, we used cluster centroids to recover the ground truth image pixel-by-pixel. The 3D occlusion Edge detection results on a sample image is shown in Figure 15. When the cluster number is set to $N = 5$ (right), most detected Edges in the ground truth image (left) are lost. We found that $N = 16$ strikes a good balance between recoverability and computation cost.

transfer to Depth from the other tasks

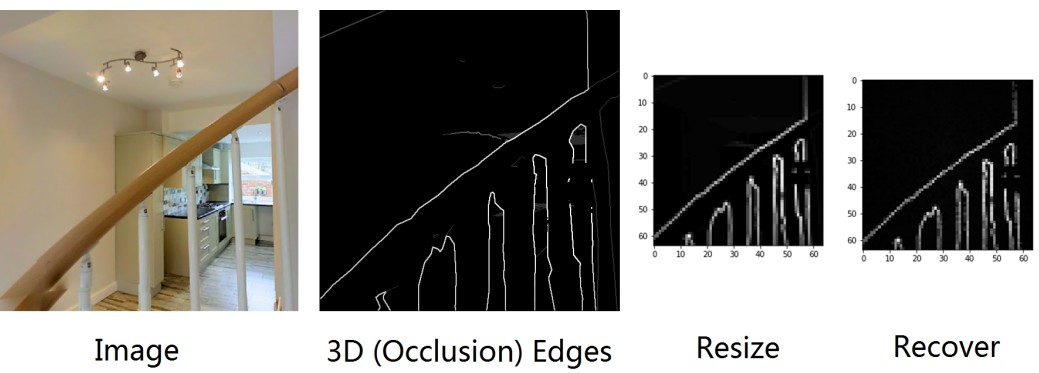

Figure 13: Comparison between source task rankings for Depth. with H-score results on the left and affinity scores Zamir et al. (2018) on the right. The Top 3 transferable source tasks in both methods are the same: Depth, Image Reshading and 3D Occlusion Edges.

| Image | 3D (Occlusion) Edges | Resize | Recover |
| --- | --- | --- | --- |

Figure 14: Quantization. Recover is done with the centroid of corresponding cluster of each pixel.

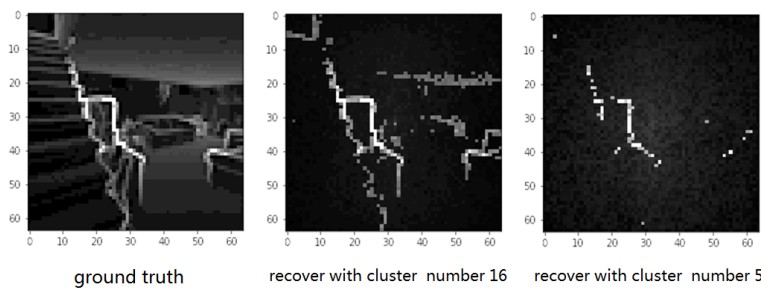

ground truth     recover with cluster number 16     recover with cluster number 5

Figure 15: Effect of quantization cluster size for 3D occlusion Edge detection.

