# OpenReview forum: "An Information-Theoretic Metric of Transferability for Task Transfer Learning"
_ICLR.cc/2019/Conference_

### Official Review · AnonReviewer3 · 2018-11-02
**In this paper, the authors considered source domain selection problem in transfer learning.**

**Rating:** 6
**Confidence:** 4

**Review:**


summary:
In this paper, the authors considered a source domain selection problem in transfer learning. Given a feature representation function f, the H-score is defined as the normalized correlation between the output f(X) and the label Y. The transferability is then measured by the ratio of H-score on the target domain and the optimal one. The authors introduced the information-theoretic and statistical meaning of the H-score. Validation of H-score was confirmed by numerical experimenters with image data.


comments:
Application of H-score to transfer learning interesting. Numerical experiments using relatively large dataset were convincing to show the validity of the proposed method. The following is some minor comments.

* Some notations and terms in section 2.1 were a bit hard to understand. e.g. what does "the error exponent corresponding to f(x)" mean? In particular, "corresponding to f(x)" was not clear for me.

* The authors showed some relationship between H-score and error of the statistical test. It would be nice to show a more direct relation between H-score and test accuracy in transfer learning. Typically, the risk in the target domain (T) is bounded above by the risk in the source domain (S) plus some dispersion between T and S as shown in the following paper:
Shen, et al., Wasserstein Distance Guided Representation Learning for Domain Adaptation, AAAI (2018).

* In the higher order transfer of numerical experiments, the concatenated features are employed. Showing a theoretical justification of such a concatenation would be nice.

---

> ### Author Response · Authors · 2018-11-23
> **Regarding feature concatenation and domain difference**
>
> The phrase “the error exponent corresponding to f(x)” actually means the error exponent of a decision region in the log ratio test parameterized by f(x). We realized the confusion and explained it with more details in the revised paper (see the latest reply).
>
> Feature concatenation is a common and efficient way to incorporate information from multiple models, especially in a neural network setting.  From a practical point of view, we assume the feature functions of individual source tasks are already computed (i.e. implemented as frozen encoder layers), before knowing about the target task. When learning the target task, concatenating different source features is the most practical thing to do to avoid information loss, with the drawback of slower speed due to increased feature dimension. Unfortunately, there is no optimality guarantee on such concatenation for a particular task.
>
> Therefore, we defined our high order task transfer learning based on this design. We treat the concatenation of n k-dimensional features as a single kn-dimensional feature f_c, and compute its H-score H(f_c) by its definition.  The transferability is computed as H(f_c)/H(f_opt) where f_opt is the optimal feature obtained by solving the HGR or Soft-HGR problem with dimension kn. This allows us to at least know how effective is feature concatenation on a standardized scale.
>
> Regarding the relationship between H-score and test accuracy in transfer learning, in this work we have not yet considered domain differences between the source and the target tasks. As stated in our definition of task transfer learning, we assume X_S and X_T are drawn from the same distribution P_X, while the joint distributions  P(X_S,Y_S) and P(X_T, Y_T) are different. In the future, we indeed plan to study domain transferability for domain adaptation problems. The reference you mentioned will be a good comparison in that study.

---

### Official Review · AnonReviewer2 · 2018-11-02
**new measure of transferability**

**Rating:** 6
**Confidence:** 3

**Review:**

The paper proposes the H score H(f), a quantity that measure the goodness of feature f(x) for predicting some target y. This heavily builds on Makur et al. (2015) who introduce information vectors, the error exponent, and the DTM matrix. The paper connects H(f) with these quantities to justify the proposal (e.g., H(f) is proportional to the error exponent (Theorem 1)). The actual transferability is measured by the ratio between H(f) and H(f_opt) where the latter can be computed using the approach of Makur et al.

The question of how to determine the relevance of a source task for a target task without learning is an important one with lots of previous work. The proposed method seems to be novel and brings many interesting ideas in Makur et al. with empirical validation. One comment is that the paper imports heavily from Makur et al. but does not make the imported definitions and results as clear as they can be. I am still unsure of what exactly the error exponent is: its definition should probably be defined in the main paper.

---

> ### Author Response · Authors · 2018-11-23
> **More definitions and background added to the revised paper**
>
> Thanks for your feedback. Please see the first comment on the changes we made for error exponents in the revised paper. We are happy to answer any further questions.

---

### Official Review · AnonReviewer1 · 2018-11-05
**Essential concepts and quantities undefined**

**Rating:** 5
**Confidence:** 3

**Review:**

The authors propose an information theoretic metric to determine a priori if a representation learned from a source task can be useful when learning another task. This is a timely topic and being able to compute such as metric between tasks upfront would be of great interest.

While the motivation and contributions of the paper are clearly stated, a number of questions remain. Central concepts like error exponent or maximal HGR correlation are mentioned in passing. The former is important to understand whether (1) makes sense and the latter is a fundamental quantity for this work. The surprising fact is that HGR is not discussed at all, nor is there any intuition provided of what this quantity represents.

While the proposed H-score is attractive when a source task has to be selected, as the denominator of the task transferability itself does not need to be computed, it is not straightforward to compute this quantity in general. The authors suggest to use the alternative conditional expectation (ACE) algorithm in order to optimize (2). ACE is not discussed in any detail (e.g., how accurate is this procedure; what are the choices and/or trade-offs if any?) and (2) is not justified. Overall, I felt section 4.2 was relatively inaccessible, but important as it indicates whether the proposed metric is of any use in practice.

Finally, I did not understand the argument that says the H-score is to be prepared over the mutual information. To me the mutual information is still the golden standard. It was also not clear why the proposed approach would not straightforwardly apply to non classification problems as suggested in the future work.

---

> ### Author Response · Authors · 2018-11-23
> **Clarifications on ACE, HGR maximal correlation and mutual information**
>
> Please see the latest comment above for the revisions on error exponents. We added more details on the ACE algorithm and its alternative in Appendix B and Section 4.2. In Appendix B, we also presented its sampling complexity.  The only trade-off parameter in ACE is the feature dimension k. A larger k generally extracts more information from the data but also increases computation cost. In practice, we choose the feature dimension based on input dimension and the constraint of computing resources.
>
> The derivation of (2) can be found in a recent paper “An Efficient Approach to Informative Feature Extraction” by Wang et al. (See https://arxiv.org/abs/1811.08979) This paper shows an alternative formulation to the ACE algorithm for computing the HGR maximal correlation. This new formulation, called soft HGR, is based on a low rank approximation of the DTM matrix B. The optimal solution then leads to Equation (2).
>
> Regarding the relationship between H-score and mutual information, we made some revision in the paper to clarify our statement. Basically, we do not attempt to replace mutual information as the golden standard for measuring the association between two random variables. Instead, we point out some practical advantage of H-score over mutual information, such as its ease of computation. This is especially the case when X is continuous, as in the image classification task. Mutual information is often computed based on binning, which has extra bias due to bin size. Other approaches either require estimating the joint distribution through kernel density estimation or using nearest neighbor approaches (Kraskov et al. 2004). On the other hand, the H-score only needs the estimation of conditional expectations, which requires less samples. And this process only takes about 10 lines of Python code.

---

### Author Response · Authors · 2018-11-23
**Thank you for your reviews!**

Thank you for your valuable feedback. Please checkout our revised paper, with changes highlighted in blue.

The common concern from all of you is the lack of explanation on error exponents.  In the revised manuscript, we added additional details in Section 2.1 and Appendix A on this subject. More information on the statistical properties of error exponents can be found in Chapters 12.7-12.9 of the book, Elements of Information Theory (Cover & Thomas 1991).

Another major concern is about HGR maximal correlation and the ACE algorithm. We added the definition of the HGR maximal correlation problem in Section 2.2. More background information on HGR as a dependence metric is discussed in Appendix B. The ACE algorithm for solving the HGR problem is listed in appendix B with a brief discussion on its convergence property.   In Section 4.2 (Efficient computation of transferability), we added more details on how to compute the optimal features that maximize H-score with respect to the target task. In particular, we make reference to the soft HGR problem (see https://arxiv.org/abs/1811.08979 and further discussion below), which is theoretically equivalent to the original HGR problem, but without the covariance constraint. The algorithm for the soft HGR problem eliminates the normalization step from the original ACE, so that we can utilize a neural network to learn the optimal features.

For other questions, we will give individual replies below each review.

---

### Author Response · Authors · 2018-11-28
**Additional experiments on NLP tasks**

We have recently performed new experiments to validate H-score as a transferability metric in NLP tasks. In particular, we tested the performance of 3 existing unsupervised word/sentence embeddings of the same dimension（fastText BOW, Globe BOW, InferSent) for 2 sentence classification tasks (CR and SUBJ) presented in [1].

We compared the H-score of the respective (embedding, task) pairs with the test accuracy  averaged over 5 trails.

Embedding Ranking based on Hscore (in parentheses)
		fastText BOW	Glove BOW	InferSent1-glove
CR		2 (0.4470) 		3 (0.4323) 	1 (0.6238)
SUBJ	1 (0.6840) 		3 (0.6701) 	2 (0.6744)

Embedding Ranking based on Test % Accuracy (in parentheses)
		fastText BOW	Glove BOW	InferSent1-glove
CR		2 (80.32) 		3 (78.99) 	1 (84.30)
SUBJ	1 (91.79) 		3 (91.05) 	2 (91.22)

As expected, embedding model rankings for each task using the two approaches are the same.  This preliminary result further demonstrates that our method can be a more efficient alternative to estimating test accuracy for learning “how transferable are features for X?”.

[1] Conneau A, Kiela D, Schwenk H, Barrault L, Bordes A. Supervised learning of universal sentence representations from natural language inference data. arXiv preprint arXiv:1705.02364. 2017 May 5.

---

### Meta-Review · Area_Chair1 · 2018-12-16
**Proposes measure for transferability with various selling points, but just ok**

**Confidence:** 4
**Recommendation:** Reject

**Metareview:**

The paper proposes an information theoretic quantity to measure the performance of transferred representations with an operational appeal, easier computation, and empirical validation.

The relation of the proposed measure to test accuracy is not considered. The operational meaning holds exactly only in the special case of linear fine tuning layers. The paper seems to import heavily from previous works.

Reviewers found it difficult to understand whether the proposed method makes sense, that the computation of relevant quantities might be difficult in general, and that the comparison with mutual information was not clear. The revision addresses these points, adding experiments and explanations. Yet, none of the reviewers gives the paper a rating beyond marginally above acceptance threshold.

All reviewers found the paper interesting and relevant, but none of them found the paper particularly strong. This is a borderline case of a sound and promising paper, which nonetheless seems to be missing a clear selling point.

I would suggest that developing the program laid out in the conclusions could make the contributions more convincing, in particular the development of more scalable algorithms and the application of the proposed measure to the design of hierarchies for transfer learning.